# Contrastive Pre-Training for Multimodal Multi-Hop Question Answering Representations

## Abstract

The multimodal multi-hop question-answering(MMQA) task is the most representative multimodal reasoning task, with its primary goal being to perform multi-step logical reasoning based on multimodal questions to obtain accurate answers. Existing MMQA methods based on large language models (LLMs) have made some progress; however, this study still faces challenges in fusing multimodal reasoning features and reasoning with multimodal multi-hop questions. To address the above issues, we propose a multimodal multi-hop representation-based contrastive pre-training (MMRCP) approach, which can effectively fuse multimodal and multi-hop question-answering features to enhance the reasoning performance of question-answering tasks. It employs two loss functions for contrastive learning training: cross-modal contrastive learning and reasoning-aware contrastive learning, which effectively obtain basic multimodal semantic features and question-answering reasoning features. Subsequently, we construct a multi-hop representation fusion module that combines multimodal reasoning features to perform lightweight adaptation for multi-hop question answering reasoning tasks. Extensive experiments on three real-world multi-hop question-answering datasets demonstrate that MMRCP outperforms multi-hop question-answering baselines by 3% and 4% in precision and error rate, respectively. MMRCP provides a promising direction for future multimodal reasoning tasks.

## 1 Introduction

With the gradual development of question answering based on large language models (LLMs), people hope that it can maintain high reasoning performance while understanding multiple modalities of information, allowing it to be applied to professional fields conveniently, such as Robot(Ni et al., 2024), Social Network, Scientific Research(Burgess et al., 2025; Wang et al., 2024) and Education(Wang et al., 2025a; Zhuang et al., 2025). In recent years, many researchers have also conducted extensive research on multimodal reasoning tasks(Dong et al., 2025; Li et al., 2025a; Wu et al., 2025). The most representative of these is the multimodal multi-hop question-answering(MMQA) research(Farahani et al., 2025; Jiang et al., 2024), which aims to conduct multiple reasoning on multimodal questions (including text and image) and gradually predict matching answers (which may also be multimodal). The primary objective of existing multimodal question-answering research is to decompose the final question progressively into logically related sub-questions and then solve them sequentially to obtain the conclusive answer. As shown in Figure1, the user attached an image when asking a question. If you want to get an accurate answer, you need to combine the image and the text question, gradually divide it into three sub-questions, and reason in sequence. Based on previous research(Ganz et al., 2024; Wang et al., 2025b), we believe that multimodal multi-hop

question answering faces two significant challenges. The first is how to integrate multimodal features more effectively and enhance the representation of semantic relationships. The second is how to establish causal relationships between semantics and improve question answering reasoning performance.

For the first challenge, some researchers have conducted relevant research. (Zhang et al., 2023) proposed using images to assist natural language understanding, thereby obtaining a universal multimodal semantic representation. (Zhang et al., 2025a) mainly focuses on the problem of regional uncertainty alignment in the process of multimodal representation learning. Some researchers have applied multimodal representation to multiple fields, such as sentiment analysis(Huan et al., 2024), person reidentification(Xiang et al., 2024), recommendation(Khan & Sisodia, 2025; Wu et al., 2024a), news detection(Li et al., 2025b; Peng et al., 2024). However, few researchers have applied it to multi-hop question answering reasoning.

In response to the second challenge, researchers have also attempted to perform related work. (Xue et al., 2024b) built an inference network based on neural symbols and causality for interpretable visual question answering. (Yang et al., 2025) proposed a language-based reasoning graph neural network for common sense question answering. (Qiu et al., 2024) performs explainable knowledge reasoning through thought chains to achieve knowledge-based visual question answering. Some researchers also combine multiple information to conduct comprehensive multi-hop reasoning question answering(Wu

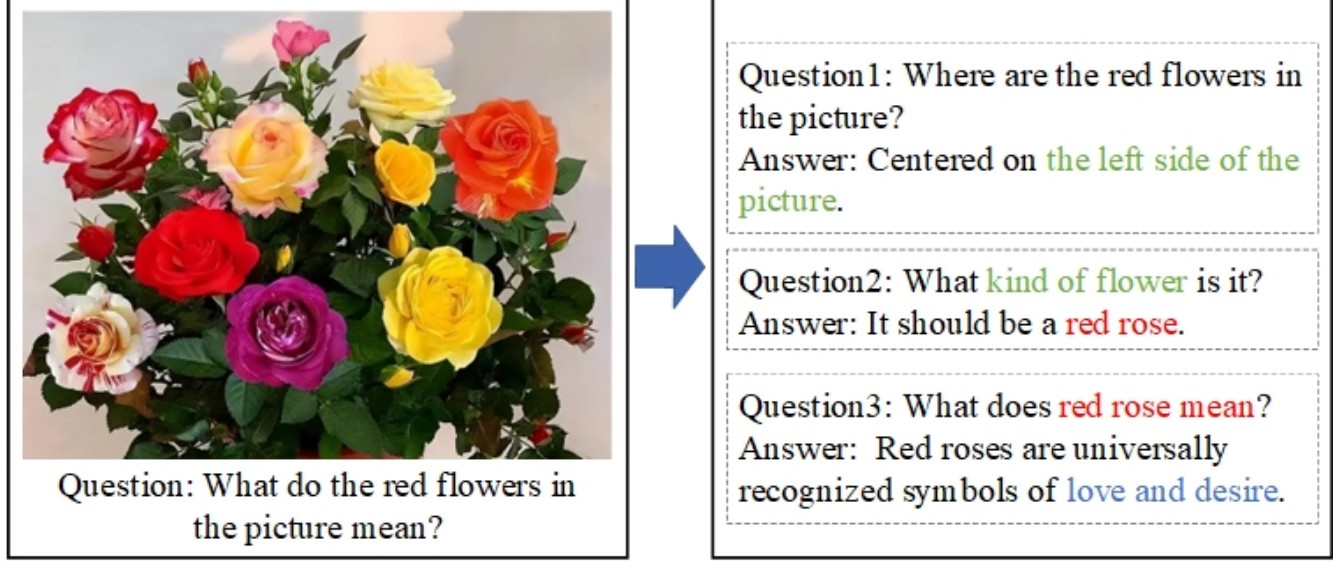

Figure 1: A multi-modal multi-hop question example.

et al., 2024b;c;d). However, most of these studies are still based on text and knowledge graphs for reasoning, and rarely involve the multimodal features of questions and answers.

To address the above challenges, we propose a multimodal multi-hop representation method based on contrastive pre-training (MMRCP) method that can effectively integrate multimodal and multi-hop question-answering features to enhance the reasoning performance of question answering tasks. For the first challenge, we employed two different loss functions for contrastive learning training: cross-modal contrastive learning and reasoning-aware contrastive learning. They can effectively obtain basic multimodal semantic features (semantic correspondence between text and image) and question-answering reasoning features (multi-hop question relations). For the second challenge, we built a multi-hop representation fusion module to combine multi-modal reasoning features and perform lightweight adaptation for multi-hop question answering reasoning tasks. Extensive experiments demonstrate that MMRCP has a significant advantage in multimodal question-answering reasoning tasks, underscoring its considerable potential in reasoning tasks. In short, the contributions of this paper are as follows:
• We originally proposed a dual contrastive learning method to obtain multimodal and multi-hop question-answering features, thereby effectively enhancing the multimodal fusion effect in the question-answering field.
• We integrated a multi-hop question-answering representation based on multimodal semantic representation and performed lightweight fine-tuning for the multi-hop question-answering task to effectively enhance multimodal question-answering reasoning performance.
• Extensive experiments on three real multi-hop question-answering datasets demonstrate that MMRCP outperforms baselines in multi-hop question-answering reasoning by 3% and 4% in precision and error rate, respectively.

## 2 RELATED WORK

### 2.1 MULTIMODAL REPRESENTATION LEARNING

The purpose of multimodal representation learning is to map data of multiple different modalities into the same vector space, laying the foundation for subsequent multimodal downstream tasks. According to its development, we divide it into three stages. **Semi-Supervised multimodal models** rely on ground-truth labeled data for fusion learning, which effectively improves the performance of specific downstream tasks, such as GW(Devillers et al., 2025), S4-Driver(Xie et al., 2025). These methods rely on expensive labeled data, and the learning results are only task-specific. **Cross-Modal pre-training methods** use the Transformer architecture to represent massive image-text pairs in different modalities, and the learning results provide a foundation for a variety of downstream tasks, for instance ViLBERT(Lu et al., 2019), VLP2MSA(Yi et al., 2024). However, it cannot provide adaptive representation for all downstream tasks. **Contrastive Learning model** establishes a unified cross-modal embedding space and performs generalization learning based on a small number of samples, which is suitable for various downstream tasks, such as BLIP-2(Li et al., 2023), InstructBLIP (Dai et al., 2023), VILA(Lin et al., 2024). The contrastive learning model is fine-tuned for different downstream tasks based on the semantic alignment of images and texts, so it provides a bright direction for existing multimodal tasks.

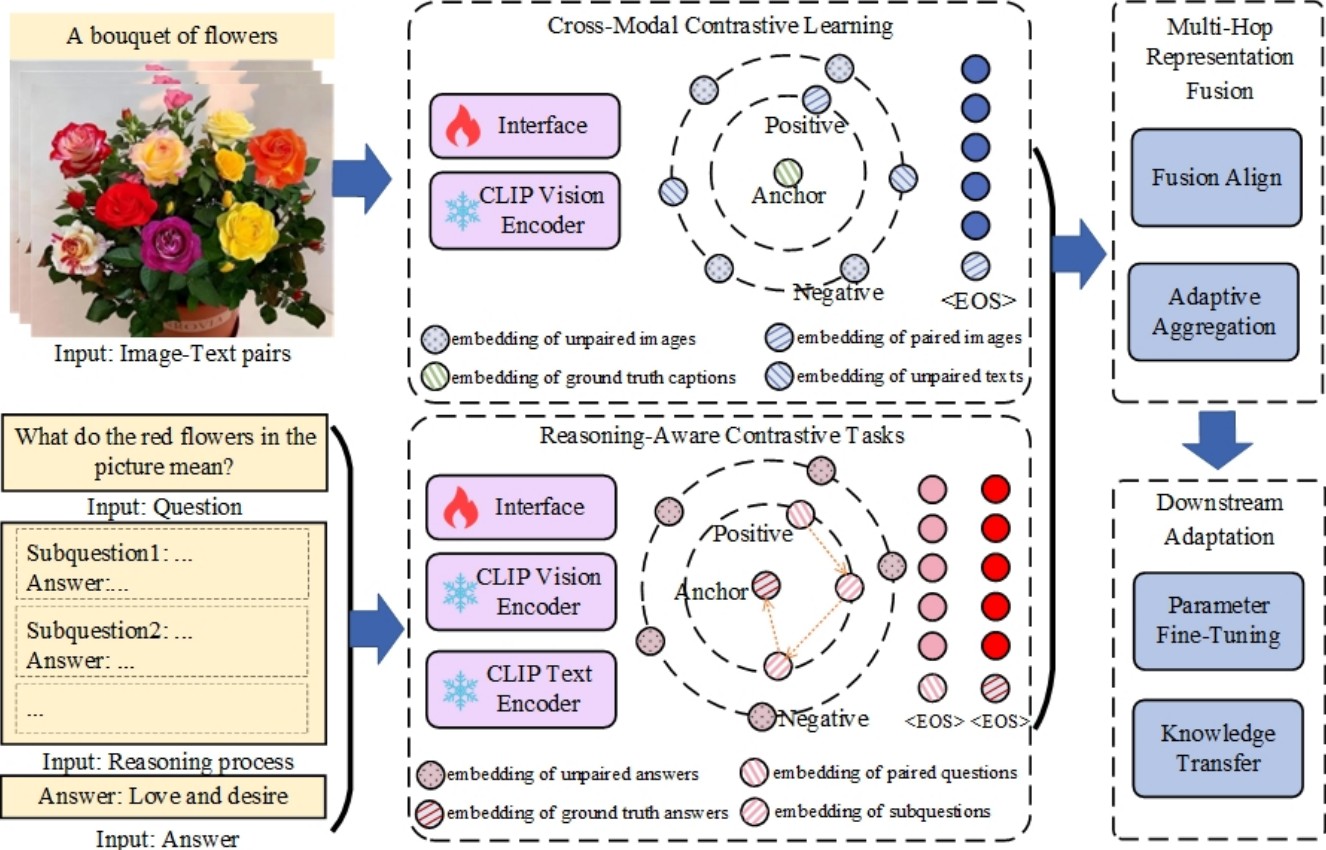

Figure 2: The overall architecture of MMRCP.

## 2.2 MULTI-HOP QUESTION ANSWERING REASONING

The research on multi-hop question answering reasoning mainly aims at solving the explanatory reasoning process of complex problems. We divide it into text multi-hop methods and multi-modal multi-hop methods according to the type of modality involved. The text multi-hop methods mainly establish question-answer relationship based on text semantic matching(Wu et al., 2023a;b) and knowledge relevance(Lu et al., 2025; Zheng et al., 2025). The multimodal multi-hop methods combined features such as text and images to complete the question-answering reasoning process collaboratively(Wu et al., 2024c;d). Most of the existing multimodal multi-hop methods are still supervised learning methods that rely on a large amount of labeled data. Therefore, multimodal contrastive learning is a promising multi-hop question answering approach.

## 3 METHOD

To overcome the challenges of multimodal multi-hop question answering reasoning representation learning, we propose a multimodal multi-hop representation method based on contrastive pre-training(MMRCP), which is a unified framework that optimizes multi-hop question answering based on aligned visual-textual semantics, thereby improving the reasoning performance of downstream tasks. It comprises four key components: Cross-Modal Contrastive Learning, Reasoning-Aware Contrastive Tasks, Multi-Hop Representation Fusion and Downstream Adaptation. The specific architecture is shown in Figure2.

### 3.1 CROSS-MODAL CONTRASTIVE LEARNING

We establish multimodal alignment through a dual-encoder contrastive paradigm. Given a training batch of image-text pairs $\{(I_i, T_i)\}_{i=1}^N$ , visual embeddings $\mathbf{v}_i$ and textual embeddings $\mathbf{t}_i$ are generated via Eq.1.

$$v_i = \frac{\phi_{vis}(I_i)}{||\phi_{vis}(I_i)||_{l_2}}, t_i = \frac{\phi_{rear}(T_i)}{||\phi_{rect}(T_i)||_{l_2}} \quad (1)$$

where $\phi_{\text{vis}}$ denotes Vision Transformer encoder, $\phi_{\text{text}}$ represents text encoder, and $\mathbf{v}_i, \mathbf{t}_i \in \mathbb{R}^d$ with $d = 768$.

The symmetric InfoNCE loss $\mathcal{L}_{\text{global}}$ optimizes cross-modal correspondence as shown in Eq.2.

$$\mathcal{L}_{\text{global}} = -\frac{1}{2N}\sum_{i=1}^{N}\left[\log\frac{e^{\mathbf{v}_i^\top \mathbf{t}_i/\tau}}{\sum\limits_{k=1}^{N}e^{\mathbf{v}_i^\top \mathbf{t}_k/\tau}} + \log\frac{e^{\mathbf{v}_i^\top \mathbf{t}_i/\tau}}{\sum\limits_{k=1}^{N}e^{\mathbf{v}_k^\top \mathbf{t}_i/\tau}}\right] \tag{2}$$

where $\mathbf{v}_i^\top \mathbf{t}_j$ computes cosine similarity, $\tau = 0.07$ controls distribution sharpness.
For fine-grained reasoning, semantic-hard negative mining constructs as can be seen from Eq.3.

$$\mathcal{N}^{(i)} = \left\{\mathbf{t}_k \mid \frac{\mathbf{t}_i^\top \mathbf{t}_k}{\|\mathbf{t}_i\|_2\|\mathbf{t}_k\|_2} > \beta\right\}_{k\neq i}^{M} \cup \mathcal{N}_{\text{rand}} \tag{3}$$

with similarity threshold $\beta = 0.8$, hard negative count $M = 3$.
The cross-modal contrastive framework establishes foundational alignment between visual and textual modalities through dual-encoder optimization. By minimizing symmetric InfoNCE loss $\mathcal{L}_{\text{global}}$ with semantic-hard negative mining $\mathcal{N}^{(i)}$, we enforce discriminative representation learning where matched pairs $(\mathbf{v}_i, \mathbf{t}_i)$ exhibit maximal cosine similarity $s(\mathbf{v}_i, \mathbf{t}_i)$, while adversarial pairs $(\mathbf{v}_i, \mathbf{t}_k \in \mathcal{N}^{(i)})$ are separated in embedding space $\mathbb{R}^d$ However, this instance-level alignment remains insufficient for modeling multi-hop reasoning chains requiring structured dependency modeling, as evidenced by $\nabla\mathcal{L}$ gradients showing limited sensitivity to compositional semantics.

To bridge this gap, we introduce reasoning-aware contrastive tasks that explicitly optimize for step-by-step inference capabilities. These tasks inject inductive biases tailored for multi-hop reasoning through three specialized mechanisms:

### 3.2 REASONING-AWARE CONTRASTIVE TASKS

The proposed contrastive tasks fundamentally redefine how multimodal representations encode compositional reasoning structures. Traditional contrastive learning methods like CLIP focus on global instance alignment, which proves inadequate for modeling the step-by-step dependency relationships inherent in multi-hop reasoning. Our framework addresses this by designing three specialized tasks that explicitly optimize for chain decomposition, path invariance, and structural consistency across modalities.

**Chain-Aware Semantic Alignment** This task establishes fine-grained correspondence between linguistic components and visual concepts along reasoning chains. Given a question $Q$ requiring multi-step inference, we first decompose it into semantic units $\{q_j\}_{j=1}^{m}$ using dependency parsing. For example, the query "Did the cellist continue playing after the violinist stopped?" decomposes into: $q_1$: "cello playing state"; $q_2$: "violin playing state"; $q_3$: "temporal sequence".
Concurrently, visual concepts $\{v_k\}$ are localized through gradient-based attention via Eq.4.

$$\alpha_k = \frac{\partial\phi_{\text{text}}(q_j)}{\partial\phi_{\text{vis}}(v_k)}, \quad v_k \in \mathcal{V}_{\text{rel}} \longleftrightarrow \alpha_k > \gamma \tag{4}$$

where $\gamma = 0.5$ filters irrelevant regions. The hierarchical contrastive loss then forces alignment between corresponding $(q_j, v_j)$ pairs via Eq.5.

$$\mathcal{L}_{\text{chain}} = -\sum_{j=1}^{m}\log\frac{\exp(\text{sim}(q_j, v_j)/\tau)}{\sum\limits_{k=1}^{K}\exp(\text{sim}(q_j, v_k)/\tau)} \tag{5}$$

**Invariant Reasoning Path Learning** Real-world reasoning must maintain consistency when partial information is missing. To simulate this challenge, we introduce stochastic perturbation $\mathcal{M}$ that randomly drops features with probability $p = 0.3$, according to Eq.6.

$$\widetilde{\mathbf{x}}_i = \mathbf{x}_i \cdot m_i, \quad m_i \sim \text{Bernoulli}(1-p) \tag{6}$$

The invariance loss then enforces representation stability, as shown in Eq.7.

$$\mathcal{L}_{\text{inv}} = \|f(\mathbf{x}) - f(\widetilde{\mathbf{x}})\|_2^2 + \lambda\max(0, \delta - \|f(\mathbf{x}) - f(\mathbf{x}_{\text{neg}})\|_2) \tag{7}$$

where $\mathbf{x}_{\text{neg}}$ denotes features from logically contradictory paths (e.g., "violinist played before cellist" vs "cello played first"), with $\delta = 1.0$ defining the margin. Crucially, this task teaches the model to distinguish between core reasoning signals and incidental features – when part of visual patches are occluded, representations must still preserve the essential inference chain.
**Cross-Modal Graph Matching** Visual scene graph $\mathcal{G}_v = (\mathcal{V}, \mathcal{E}_v)$ where edges $\mathcal{E}_v$ encode spatial/temporal relations. **Linguistic dependency graph** $\mathcal{G}_l = (\mathcal{Q}, \mathcal{E}_l)$ with $\mathcal{E}_l$ representing syntactic dependencies. Graph neural networks project these structures into aligned embeddings via Eq.8.

$$\mathbf{h}_v = \text{GNN}_{\theta_v}(\mathcal{G}_v), \quad \mathbf{h}_l = \text{GNN}_{\theta_l}(\mathcal{G}_l) \tag{8}$$

The graph contrastive loss then optimizes structural isomorphism, following from Eq.9.

$$\mathcal{L}_{\text{graph}} = -\log \frac{\exp(\mathbf{h}_v^\top \mathbf{h}_l / \tau)}{\sum\limits_{k=1}^{B} \exp(\mathbf{h}_v^\top \mathbf{h}_l^{(k)} / \tau)} \tag{9}$$

where $\mathbf{h}_l^{(k)}$ are negative graph embeddings from mismatched questions. This approach uniquely preserves relational seman­tics–for instance, the "temporal-follow" relation between instruments remains consistent whether expressed visually or lin­guistically.

Synergistic Integration The joint optimization framework combines these tasks through adaptive weighting, based on Eq.10.

$$\mathcal{L}_{\text{reason}} = \sum_{i=1}^{3} \lambda_i \mathcal{L}_i, \quad \lambda_i = \sigma(\mathbf{w}^\top \mathbf{f}_i) \tag{10}$$

with $\sigma$ denoting the sigmoid function and $\mathbf{f}_i$ task-specific features. This dynamic balancing allows the model to emphasize chain alignment for attribute-heavy queries while prioritizing graph matching for relation-centric questions – a flexibility critical for handling diverse multi-hop scenarios.

The reasoning-aware contrastive framework fundamentally advances multimodal representation learning by embedding explicit structural biases for compositional inference. Through the chain-aligned semantic optimization $\mathcal{L}_{\text{chain}}$ , we establish step-wise correspondence between linguistic sub-queries $\{q_j\}$ and visual concepts $\{v_k\}$, overcoming the limitation of global alignment in modeling partial-order dependencies along reasoning paths $\mathcal{P}$. Simultaneously, the path invariance mechanism $\mathcal{L}_{\text{inv}}$ ensures logical consistency under stochastic perturbations $\widetilde{\mathbf{x}} = \mathbf{x} \odot \mathbf{m}$ with masking probability $p = 0.3$, maintaining core inference integrity despite significant feature occlusion. Crucially, the cross-modal graph alignment $\mathcal{L}_{\text{graph}}$ preserves relational semantics through isomorphic mapping between visual scene graphs $\mathcal{G}_v$ and linguistic dependency structures $\mathcal{G}_l$ , enabling precise encoding of relational predicates $\exists!$.

These synergistic tasks generate heterogeneous yet complementary representations: $\mathbf{H}_{\text{chain}} \in \mathbb{R}^{m \times d}$ captures sequen­tial reasoning steps, $\mathbf{H}_{\text{inv}} \in \mathbb{R}^d$ embodies perturbation-robust logical cores, and $\mathbf{H}_{\text{graph}} \in \mathbb{R}^{|\mathcal{E}| \times d}$ stores cross-modal relational constraints. However, effective multi-hop question answering demands context-aware synthesis of these distinct representation types–temporal questions require emphasis on chain progression $\partial \mathcal{P} / \partial t$ , spatial queries prioritize relational graph embeddings $\mathcal{G}$, while existence verification depends on invariant features $\nabla_{\mathcal{P}} f$.

This necessity motivates our Multi-Hop Representation Fusion module, which dynamically consolidates the heteroge­neous representations through attention-based gating conditioned on question semantics $f_q$. The fusion mechanism adaptively weights $\mathbf{H}^{(i)}$ components according to query-specific requirements, enabling coherent integration of temporal, spatial, and relational evidence for complex reasoning scenarios where multifaceted clues must be holistically synthesized into unified reasoning pathways $\Gamma$.

## 3.3 MULTI-HOP REPRESENTATION FUSION

**Fusion Mechanism Formulation.** Given the decomposed reasoning representations - sequential chain embeddings $\mathbf{H}_{\text{chain}} \in \mathbb{R}^{m \times d}$, invariant path features $\mathbf{H}_{\text{inv}} \in \mathbb{R}^d$, and relational graph embeddings $\mathbf{H}_{\text{graph}} \in \mathbb{R}^{|\mathcal{E}| \times d}$ - we compute attention weights conditioned on question semantics $\mathbf{q} \in \mathbb{R}^d$, as derived from Eq.11.

$$\alpha_i = \text{softmax}\left(\mathbf{W}_a^\top \tanh(\mathbf{W}_h \mathbf{H}^{(i)} + \mathbf{W}_q \mathbf{q})\right) \tag{11}$$

where $\{\mathbf{H}^{(i)}\} = \{\mathbf{H}_{\text{chain}}, \mathbf{H}_{\text{inv}}, \mathbf{H}_{\text{graph}}\}$. This attention mechanism enables context-dependent weighting: temporal queries amplify $\mathbf{H}_{\text{chain}}$, spatial questions prioritize $\mathbf{H}_{\text{graph}}$, while verification tasks focus on $\mathbf{H}_{\text{inv}}$.

**Adaptive Feature Aggregation** The fused representation $\mathbf{h}_{\text{fused}}$ combines modality-specific features through residual gating via Eq.12.

$$\mathbf{h}_{\text{fused}} = \sum_{i=1}^{3} \alpha_i \cdot \left(\mathbf{W}_i \mathbf{H}^{(i)}\right) + \beta \cdot \mathcal{G}\left(\oplus_{i=1}^{3} \mathbf{H}^{(i)}\right) \tag{12}$$

Here $\oplus$ denotes concatenation, $\mathcal{G}(\cdot)$ is a two-layer Gated Recurrent Unit (GRU) with hidden dimension $d/2$, and $\beta$ is a learnable scaling parameter initialized at 0.5.

**Dimensionality Harmonization** To resolve dimensional conflicts between $\mathbf{H}_{\text{chain}}$ (matrix) and $\mathbf{H}_{\text{inv}}$ (vector), we apply position-wise pooling, as described by Eq.13.

$$\widetilde{\mathbf{H}}_{\text{chain}} = \max_{j=1}^{m}\left(\mathbf{H}_{\text{chain}}[j, :]\right), \quad \widetilde{\mathbf{H}}_{\text{graph}} = \frac{1}{|\mathcal{E}|} \sum_{k=1}^{|\mathcal{E}|} \mathbf{H}_{\text{graph}}[k, :] \tag{13}$$

This transformation ensures dimensional homogeneity $\widetilde{\mathbf{H}}^{(i)} \in \mathbb{R}^d$ for fusion while preserving critical information through max-pooling (capturing dominant reasoning steps) and average-pooling (maintaining relational semantics).

The multi-hop representation fusion module achieves context-aware integration of heterogeneous reasoning features through dual-path gating architecture. By computing attention weights $\alpha_i$ conditioned on question semantics $\mathbf{q}$, the mechanism dynamically prioritizes $\mathbf{H}_{\text{chain}}$for temporal sequencing queries, $\mathbf{H}_{\text{graph}}$ for spatial relationship questions, and $\mathbf{H}_{\text{inv}}$ for existence verification tasks. The residual gating formulation $\mathbf{h}_{\text{fused}} = \sum \alpha_i \mathbf{W}_i \mathbf{H}^{(i)} + \beta \cdot \mathcal{G}(\oplus \mathbf{H}^{(i)})$ preserves both task-specific semantics and cross-representation dependencies, while dimensional harmonization via max-pooling and $\mathbb{E}[\cdot]$ pooling ensures feature compatibility. Crucially, this yields a unified reasoning representation $\mathbf{h}_{\text{fused}} \in \mathbb{R}^d$ exhibiting three fundamental properties: compositionality (decomposed step encoding), robustness (perturbation invariance), and relational awareness (structural dependency mapping).

The compact yet expressive nature of $\mathbf{h}_{\text{fused}}$ now necessitates minimalist downstream adaptation—a strategic imperative for efficient deployment where computational overhead must be optimized without compromising inference capability. This seamlessly motivates our lightweight adaptation framework, which transforms this unified representation into answer predictions through parameter-efficient operations. By maintaining maximal backbone parameter freezing while introducing only essential task-specific layers, the subsequent adaptation module demonstrates how distilled reasoning semantics can be directly operationalized for end-to-end question answering with minimal architectural expansion.

### 3.4 DOWNSTREAM ADAPTATION

**Lightweight Projection Architecture.** The transformation operates through sequential nonlinear projections as expressed in Eq.14.

$$
\begin{aligned}
\mathbf{z} &= \text{GELU}\left(\mathbf{W}_1 \mathbf{h}_{\text{fused}} + \mathbf{b}_1\right) \\
\widehat{\mathbf{y}} &= \text{softmax}\left(\mathbf{W}_2 \mathbf{z} + \mathbf{b}_2\right)
\end{aligned}
\tag{14}
$$

Here the hidden dimension $d_h = \lfloor d/2 \rfloor$ creates a computational bottleneck that reduces parameters by 50% compared to full-dimensional projection. The optimization minimizes cross-entropy loss $\mathcal{L}_{\text{CE}}$ defined as Eq.15.

$$
\mathcal{L}_{\text{CE}} = -\sum_{i=1}^{N} \sum_{c=1}^{C} y_{i,c} \log(\hat{y}_{i,c})
\tag{15}
$$

where $y_{i,c} \in \{0, 1\}$ denotes ground-truth labels.

**Knowledge Preservation Mechanism** To prevent catastrophic forgetting of pre-trained representations, elastic weight consolidation imposes soft constraints on critical parameters, as indicated in Eq.16.

$$
\mathcal{L}_{\text{EWC}} = \sum_i \frac{\lambda}{2} F_i (\theta_i - \theta_{i,0}^*)^2
\tag{16}
$$

where $F_i$ represents diagonal elements of the Fisher information matrix, $\theta_{i,0}^*$ denotes original pre-trained weights, and $\lambda = 10^3$ controls constraint strength. The composite loss $\mathcal{L} = \mathcal{L}_{\text{CE}} + \mathcal{L}_{\text{EWC}}$ balances task adaptation with representation stability.

Zero-Shot Transfer Capability The modular architecture supports seamless adaptation to novel question-answering formats through dynamic re-initialization of the final projection layer. When encountering new answer vocabularies, only the weight matrix $\mathbf{W}_2 \in \mathbb{R}^{d_h \times C_{\text{new}}}$ requires updating while retaining all upstream components. This flexibility enables efficient deployment across diverse QA benchmarks without full model retraining.

The adaptation design demonstrates how distilled multimodal reasoning representations can be operationalized through minimal computational overhead. By maintaining maximal parameter freezing and employing strategic regularization, the framework achieves effective knowledge transfer while preserving the compositional reasoning capabilities learned during contrastive pretraining.

### 3.5 OVERALL ALGORITHM

The complete workflow is formalized in Appendix B, which integrates all components into a coherent computational procedure. This algorithm provides a comprehensive blueprint of the method's execution flow, beginning with multimodal contrastive pre-training (Lines 2-8), progressing through reasoning-specific optimization (Lines 10-16), and concluding with representation fusion and task adaptation stages (Lines 18-24) that generate final answer predictions. The implementation preserves all formally defined mathematical operations while abstracting implementation details into an executable computational structure.

## 4 EXPERIMENTS

### 4.1 DATASETS

In the pre-training contrastive learning stage of LLaVA 1.5(Liu et al., 2024), we use LAION, Conceptual 12M, COYO-700M and DFN2B datasets to train model-based text and image semantic alignment. Since the cost of pre-training the complete dataset is too high, we randomly sample 5% of the four datasets to construct the pre-training dataset, which can reduce the cost of pre-training and avoid the impact of dataset specificity. The detailed information of pre-training datasets is shown in Table 3 (See Appendix A). To verify the effectiveness of MMRCP in multi-hop reasoning, we selected three multimodal question-answering datasets from Stack Exchange[1] , the largest question-answering community in the world. The detailed information is shown in Table 4 (See Appendix A).

### 4.2 EVALUATION METRICS

To effectively evaluate multi-hop question answering performance, we employ three metrics: Precision_K, MSE_K, and NRMSE_K.

**Precision_K** measures the accuracy of top-K prediction results, serving as the standard evaluation metric for answer recommendation systems, as shown in Eq.17.

$$Precision\_K = \frac{TP\_K}{TP\_K + FP\_K} \tag{17}$$

where $TP\_K$ is the number of True Positives answer in the top K results, and $FP\_K$ means the count of False Positive answers in the top K result.

**MSE_K** (Mean Squared Error) quantifies the average squared difference between the top-K predicted and actual answers, as illustrated in Eq.18.

$$MSE\_K = \frac{1}{K} \sum_{i=1}^{K} (\hat{y}_i - y_i)^2 = \frac{1}{K} \sum_{i=1}^{K} (\text{ Distance } (\hat{y}_i, y_i))^2 \tag{18}$$

Where $\hat{y}_i$ is the $i$ th prediction result, and $y_i$ represents the corresponding true result. Its value range is 0 to infinity. When the predicted results match the actual results, it equals 0, which is the optimal model.

**NRMSE_K** (Normalized Root Mean Squared Error) represents the normalized deviation between predicted and true answers, as outlined in Eq.19.

$$NRMSE\_K = \frac{\sqrt{MSE\_K}}{\max_{\forall i,j \in [1,K]} (\text{ Distance } (y_i - y_j))} \tag{19}$$

### 4.3 IMPLEMENTATION DETAILS

This method is mainly implemented in Python. The local physical environment is an Intel i7 processor with a memory size of 32 GB. The pre-training process uses a rented AutoDL server with configurations such as RTX 4090, RTX5090, etc. In practical applications, parameters can be adjusted according to the target scenario. In the experimental evaluation phase, we repeated the same test 3 times and used the average of the results as a comparison benchmark to avoid the impact of randomness of data partitioning.

### 4.4 BASELINES

To verify the performance of MMRCP in multi-hop question-answering reasoning, we designed three sets of comparative experiments. The first set selected a group of recent multimodal question-answering comparison baselines and directly reasoned to get the answers. The second set selected some popular multimodal text-image embedding methods, embedded the given multimodal question-answer pair, and then used method MHCQA(Wu et al., 2024d) to reason about the question-answer relationship. The third set selected some multimodal pre-training methods to represent the questions, and then used method MHCQA to reason about the answer.

**Group One** includes II-MMR(Kil et al., 2024), QC-MHM(Xue et al., 2024a), and KEDKG(Lu et al., 2025).

**Group Two** includes PolCLIP(Yang et al., 2024), MLA(Zhang et al., 2024), NoteLLM-2(Zhang et al., 2025b), and DSL-MRS(Khan & Sisodia, 2025).

**Group Three** includes CLIP(Radford et al., 2021), BLIP(Li et al., 2022), BLIP2(Li et al., 2023), LLaVA(Liu et al., 2023), LLaVA-1.5(Liu et al., 2024), PROMISE(Wu et al., 2024a), AutoML(Moharil et al., 2024), and miniGPT-4(Zhu et al., 2024).

---

[1] https://archive.org/details/stackexchange

| Method | Stack Overflow | | | Mathematics | | | Super User | | |
|---|---|---|---|---|---|---|---|---|---|
| | Prec_K | MSE_K | NRMSE_K | Prec_K | MSE_K | NRMSE_K | Prec_K | MSE_K | NRMSE_K |
| II-MMR | 0.52 | 0.64 | 0.63 | 0.39 | 0.67 | 0.65 | 0.41 | 0.66 | 0.64 |
| QC-MHM | 0.51 | 0.63 | 0.61 | 0.38 | 0.66 | 0.64 | 0.4 | 0.65 | 0.63 |
| KEDKG | 0.53 | 0.62 | 0.61 | 0.41 | 0.65 | 0.64 | 0.42 | 0.64 | 0.62 |
| PolCLIP | 0.57 | 0.68 | 0.66 | 0.45 | 0.71 | 0.68 | 0.46 | 0.7 | 0.67 |
| MLA | 0.61 | 0.66 | 0.66 | 0.49 | 0.7 | 0.69 | 0.51 | 0.68 | 0.68 |
| NoteLLM-2 | 0.58 | 0.65 | 0.64 | 0.46 | 0.68 | 0.67 | 0.46 | 0.67 | 0.66 |
| DSL-MRS | 0.64 | 0.64 | 0.64 | 0.53 | 0.67 | 0.66 | 0.53 | 0.66 | 0.65 |
| CLIP | 0.53 | 0.75 | 0.74 | 0.41 | 0.79 | 0.77 | 0.43 | 0.77 | 0.75 |
| BLIP | 0.56 | 0.73 | 0.71 | 0.41 | 0.76 | 0.73 | 0.45 | 0.75 | 0.73 |
| BLIP2 | 0.57 | 0.73 | 0.73 | 0.46 | 0.76 | 0.76 | 0.47 | 0.75 | 0.74 |
| LLaVA | 0.61 | 0.68 | 0.66 | 0.47 | 0.71 | 0.68 | 0.5 | 0.7 | 0.68 |
| LLaVA-1.5 | 0.67 | 0.62 | 0.6 | 0.51 | 0.64 | 0.62 | 0.56 | 0.64 | 0.62 |
| PROMISE | 0.61 | 0.69 | 0.63 | 0.46 | 0.71 | 0.69 | 0.47 | 0.68 | 0.68 |
| AutoML | 0.56 | 0.73 | 0.69 | 0.47 | 0.72 | 0.68 | 0.49 | 0.69 | 0.71 |
| miniGPT-4 | 0.63 | 0.64 | 0.64 | 0.51 | 0.66 | 0.66 | 0.51 | 0.66 | 0.66 |
| **MMRCP** | **0.68** | **0.61** | **0.59** | **0.53** | **0.64** | **0.62** | **0.56** | **0.63** | **0.61** |

Table 1: Experimental results of multi-hop question-answering reasoning.

## 4.5 MAIN RESULTS

To verify the multi-hop question answering reasoning effect of MMRCP, we conducted comparative experiments with baseline methods on three datasets. The experimental results are shown in Table 1. By comparing the experimental results from the following three aspects, we get some promising inspirations. In multi-hop QA reasoning, the pretraining group achieved superior average performance owing to semantic pretraining on large-scale text-image pairs, which enhanced text comprehension and cross-modal alignment. MMRCP led this group due to its pretraining with multi-hop QA contrastive learning, enabling precise image-text semantic relation understanding. Multimodal embedding methods consistently outperformed multimodal reasoning approaches, as embedding-based techniques prioritize text-image representation learning that captures semantic relationships. Conversely, most reasoning methods exhibit limited generalizability, being optimized for specific downstream tasks and consequently demonstrating unstable performance across diverse domains and datasets.

In terms of multi-hop dataset properties, the baseline methods generally perform better on Stack Overflow, the key factor being that it has a larger Q&A training sample and the semantic relevance of text and images is higher. The baseline methods perform better than Mathematics on the smaller Super User, mainly because the question-and-answer images in Mathematics are mainly formulas or equations, which becomes the biggest obstacle to the semantic matching of text and images.

Multimodal pretraining (Group Three) significantly enhanced question-answering (QA) reasoning, primarily attributable to large-scale text-image datasets enabling effective cross-modal alignment. While CLIP established foundational alignment capabilities, it depended on annotated data while neglecting image-text semantic relationships. BLIP mitigated alignment noise but overlooked relationship reasoning. BLIP2 integrated multiple models for visual-linguistic representation learning, yet exhibited weak QA reasoning. Subsequent models (miniGPT4, LLaVA series) augmented BLIP2 with LLM-generated visual instructions to optimize representation learning, but lacked reasoning-focused QA data for performance enhancement. PROMISE leveraged knowledge graphs for improved matching but faced domain constraints limiting QA applicability. AutoML prioritized generic multimodal representation learning but underperformed in multi-hop reasoning. In contrast, MM-RCP adopted visual-language representations from miniGPT4/LLaVA while specifically optimizing reasoning with multi-hop QA datasets, achieving state-of-the-art performance. Therefore, optimal multimodal multi-hop QA reasoning requires synergistic integration of: (1) robust cross-modal semantic representation, and (2) task-specific reasoning optimization.

## 4.6 ABLATION STUDIES

The MMRCP consists of Cross-Modal Contrastive Learning (CMCL), Reasoning-Aware Contras-tive Tasks (RACT), Multi-Hop Representation Fusion (MHRF) and Downstream Adaptation (DA). To verify the importance of each component module to MMRCP, we conducted the following ablation experiment.

By comparing the experimental results, we can draw the following conclusions: (1) All four modules have a positive effect on the overall performance; (2) The CMCL module has the greatest impact on the overall performance because it is the basis of multimodal semantic representation; (3) The MHRF module has the least impact on the overall performance,

| **Methods** | Stack Overflow | | |
| --- | --- | --- | --- |
| | Prec_K | MSE_K | NRMSE_K |
| MMRCP w/o. CMCL | 0.35 | 0.84 | 0.68 |
| MMRCP w/o. RACT | 0.46 | 0.71 | 0.66 |
| MMRCP w/o. MHRF | 0.62 | 0.73 | 0.67 |
| MMRCP w/o. DA | 0.58 | 0.68 | 0.63 |
| **MMRCP** | **0.68** | **0.61** | **0.59** |

(a) Ablation study of MMRCP on Stack Overflow

| $\eta_{\text{pretrain}}/\eta_{\text{finetune}}$ | Stack Overflow | | |
| --- | --- | --- | --- |
| | Prec_K | MSE_K | NRMSE_K |
| 0.1/0.9 | 0.59 | 0.68 | 0.64 |
| 0.3/0.7 | 0.61 | 0.65 | 0.62 |
| 0.5/0.5 | 0.65 | 0.62 | 0.61 |
| 0.7/0.3 | **0.68** | **0.61** | **0.59** |
| 0.9/0.1 | 0.66 | 0.63 | 0.62 |

(b) Learning rate experiment on Stack Overflow.

| **Temperature $\tau$** | Stack Overflow | | |
| --- | --- | --- | --- |
| | Prec_K | MSE_K | NRMSE_K |
| 0.2 | 0.66 | 0.68 | 0.61 |
| 0.4 | **0.68** | **0.61** | **0.59** |
| 0.6 | 0.65 | 0.67 | 0.63 |
| 0.8 | 0.63 | 0.71 | 0.65 |
| 1.0 | 0.57 | 0.84 | 0.68 |

(c) Temperature experiment on Stack Overflow.

| **Similarity threshold $\beta$** | Stack Overflow | | |
| --- | --- | --- | --- |
| | Prec_K | MSE_K | NRMSE_K |
| 0.2 | 0.84 | 0.94 | 0.71 |
| 0.4 | 0.78 | 0.88 | 0.68 |
| 0.6 | 0.71 | 0.78 | 0.63 |
| 0.8 | **0.68** | **0.61** | **0.59** |
| 1.0 | 0.61 | 0.59 | 0.57 |

(d) Similarity threshold on Stack Overflow.

Table 2: Ablation and hyperparameter experiments of MMRCP on Stack Overflow.

probably because the early multimodal and reasoning contrastive learning has laid the foundation for multi-hop reasoning.

Meanwhile, we analyze the impact of all hyperparameters on the performance of MMRCP.

**Learning rates** $\eta_{\text{pretrain}}, \eta_{\text{finetune}}$: We set up several sets of learning rate experiments as shown in Table 2b. The experimental results show that MMRCP has better effect when the pre-training learning rate and the tuning learning rate are 0.7 and 0.3.

**Temperature** $\tau$: Table 2c shows the results of several groups of experiments we selected, among which the best effect is when the temperature coefficient is 0.4.

**Similarity threshold** $\beta$: Table 2d shows the results of our comparative experiments, where MMRCP achieves better results when the similarity threshold is 0.8.

## 5 CONCLUSION

This paper focuses on the challenge of multi-hop and multimodal question answering reasoning. It proposes a multimodal multi-hop representation method based on contrastive pre-training (MMRCP) to improve the question answering reasoning effect. Based on a large amount of text-image pre-training, MMRCP integrates multi-hop question-answering contrast features, so its representation learning results are more effectively adapted to multi-hop question-answering reasoning tasks. Experimental results show that MMRCP improves the precision and error rate of baselines in multi-hop reasoning by about 3% and 4%. Based on it, complex multimodal reasoning tasks have become a promising research direction in the future, such as project architecture design, complex agent planning, etc.

## 6 LLM USAGE

During the preparation of this manuscript, the authors employed LLM for language polishing and grammatical corrections. Following the use of this tool, the authors thoroughly reviewed and revised the content as necessary and accept full responsibility for the entire publication.

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

## A  DATASET

| Datasets | Size | Text-Image Pairs |
|---|---|---|
| LAION(Schuhmann et al., 2022) | 10T | 400 M |
| Conceptual 12M(Changpinyo et al., 2021) | 2.12G | 12 M |
| COYO-700M | 4T | 747 M |
| DFN2B(Fang et al., 2024) | 4.5T | 12.8 B |

Table 3: Datasets for multimodal contrastive pre-training.

| Datasets | Q/A | Types |
|---|---|---|
| Stack Overflow [2] | 24M/36M | Text, Image |
| Mathematics [3] | 1.7M/2.2M | Text, Image |
| Super User [4] | 514K/747K | Text, Image |

Table 4: Multimodal Multi-hop Question Answering Datasets.

# B  ALGORITHM

---

**Algorithm 1: Multimodal Contrastive Pre-training for Multi-hop QA**

---

**Input:** Image-text pairs $\{(I_i, T_i)\}$; Multi-hop QA samples $\{(Q_j, A_j)\}$
**Parameter:** Learning rates $\eta_{\text{pretrain}}, \eta_{\text{finetune}}$; Temperature $\tau$; Similarity threshold $\beta$; Loss weights $\lambda_{\text{global}}, \lambda_{\text{chain}}, \lambda_{\text{inv}}, \lambda_{\text{graph}}$
**Output:** Fine-tuned QA model $\phi_{\text{vis}}, \phi_{\text{text}}$
1: Initialize encoders $\Theta^*$.
2: **//Cross-Modal Contrastive Learning**
3: while not converged do
4:     $\mathbf{v}_i \leftarrow \phi_{\text{vis}}(I_i)$
5:     $\mathbf{t}_i \leftarrow \phi_{\text{text}}(T_i)$
6:     $\mathcal{L}_{\text{global}} \leftarrow -\log \frac{\exp(s(\mathbf{v}_i, \mathbf{t}_i)/\tau)}{\sum_k \exp(s(\mathbf{v}_i, \mathbf{t}_k)/\tau)}$
7:     Update $\phi_{\text{vis}}, \phi_{\text{text}}$ with $\nabla \mathcal{L}_{\text{global}}$
8: end while
9:
10: **// Reasoning-Aware Contrastive Tasks**
11: while not converged do
12:     $\{q_j\}_{j=1}^m \leftarrow \text{Decompose}(Q)$
13:     $\mathcal{L}_{\text{chain}} \leftarrow -\sum_j \log \frac{\exp(s(q_j, v_j)/\tau)}{\sum_k \exp(s(q_j, v_k)/\tau)}$
14:     $\mathcal{L} \leftarrow \lambda_{\text{global}} \mathcal{L}_{\text{global}} + \lambda_{\text{chain}} \mathcal{L}_{\text{chain}}$
15:     Update $\Theta$ with $\nabla \mathcal{L}$
16: end while
17:
18: **// Multi-Hop Fusion & Adaptation**
19: $\mathbf{h}_{\text{fused}} \leftarrow \sum_i \alpha_i \mathbf{H}^{(i)}$
20: while not converged do
21:     $\hat{y} \leftarrow \text{softmax}(\mathbf{W}_2 \text{GELU}(\mathbf{W}_1 \mathbf{h}_{\text{fused}}))$
22:     $\mathcal{L}_{\text{CE}} \leftarrow -\sum_c y_c \log(\hat{y}_c)$
23:     Update MLP with $\nabla \mathcal{L}_{\text{CE}}$
24: end while
25: return $\Theta^*$

---

[2] https://stackoverflow.com/
[3] https://math.stackexchange.com/
[4] https://superuser.com/

