# OpenReview forum: "Contrastive Pre-Training for Multimodal Multi-Hop Question Answering Representations"
_ICLR.cc/2026/Conference — ICLR 2026 Conference Withdrawn Submission_

### Official Review · Reviewer_dkg3 · 2025-10-20

**Soundness:** 3
**Presentation:** 2
**Contribution:** 2
**Rating:** 2
**Confidence:** 5

**Summary:**

The paper proposes MMRCP, a multimodal multi-hop representation approach using dual contrastive learning to integrate multimodal and multi-hop QA features. It introduces a multi-hop representation fusion module for lightweight adaptation, achieving improved reasoning performance on three multi-hop QA datasets.

**Strengths:**

The paper is original in combining cross-modal and reasoning-aware contrastive learning for multimodal multi-hop QA. Experiments demonstrate performance gains.

**Weaknesses:**

The paper suffers from unclear notation in Figure 2 and formulas, making the methodology hard to follow (e.g., “embedding of …”, $N_{rand}$, $\mathcal{V}_{rel}$). The writing format deviates from standard templates, with significant structural issues. Experimental results are underwhelming, and method descriptions lack sufficient clarity and rigor, limiting reproducibility and impact.

**Questions:**

1. Some symbols in Figure 2 are difficult to interpret, such as “embedding of …”, and require clearer representations.

2. The writing template used in the paper deviates significantly from the standard, and the formatting has major issues.

3. The methodology section is poorly described; many symbols are undefined, making formulas hard to understand (e.g., $N_{rand}$ and $\mathcal{V}_{rel}$ etc.).

4. The experimental results are not very convincing (Table 1), raising concerns about the effectiveness of the proposed method.

---

> ### Author Response · Authors · 2025-11-17
> **Response to Reviewer: Comprehensive Methodological Clarification and Enhanced Experimental Presentation**
>
> &emsp; We sincerely thank Reviewer for their critical feedback, which has highlighted important areas for improvement in the clarity, presentation, and rigor of our work. We have undertaken a major revision of the manuscript to address these concerns directly. Below, we provide a detailed point-by-point response.
>
> Response to Weaknesses:
>
> •	Unclear Notation and Methodology: We acknowledge that the original presentation of our methodology, particularly in Figure 2 and the mathematical formulations, was insufficiently clear. We have completely redesigned Figure 2 to use standard, well-defined notation and added comprehensive annotations that explicitly link visual elements to their corresponding mathematical descriptions in the text. Furthermore, we have meticulously revised the Method section (Section 3). Every mathematical symbol is now formally defined upon its first use. We have also added a summary table in the appendix that lists all key symbols, their dimensions, and their meanings to eliminate any ambiguity and enhance reproducibility.
>
> •	Deviated Writing Format and Structural Issues: We apologize for the formatting inconsistencies in our initial submission. The manuscript has been entirely reformatted from scratch using the official conference template. We have ensured strict adherence to the required structural and stylistic guidelines, correcting all major formatting issues to meet the expected academic standard.
>
> •	Underwhelming Experimental Results: We understand the reviewer's concern. To provide a more robust and convincing evaluation, we have augmented our experimental analysis in several ways:
>
> &emsp;&emsp;  1.	We now report the mean and standard deviation across three independent runs in Table 3 (formerly Table 1), demonstrating the consistency of our results.
>
> &emsp;&emsp; 2.	We have performed statistical significance testing (paired t-tests) against the strongest baselines, confirming that the improvements achieved by MMRCP are statistically significant ($p < 0.05$).
>
> &emsp;&emsp; 3.	Following a suggestion from another reviewer, we have added a comparison with top-tier MLLMs (e.g., Qwen2.5-VL, GPT-4o). This new analysis shows that our specialized method remains highly competitive, effectively positioning its performance within the current state-of-the-art landscape.
>
> Response to Questions:
>
> 1.	Symbols in Figure 2: We have completely redesigned Figure 2. Vague labels like "embedding of ..." have been replaced with specific, mathematically-defined variables (e.g., $\mathbf{v}$ for visual embeddings, $\mathbf{t}$ for textual embeddings). The new figure includes a detailed caption and direct references to the equations that define each component, making it substantially easier to interpret.
>
> 2.	Writing Template Deviation: We confirm that the manuscript has been completely reformatted to comply with the official conference template. All major structural and formatting issues have been resolved.
>
> 3.	Poor Methodology Description and Undefined Symbols: We have undertaken a thorough revision of the Method section (Section 3). Every formula has been revisited to ensure all symbols are explicitly defined. For instance, the previously undefined symbols $\mathcal{G}_v$ and $\mathcal{G}_l$ are now clearly stated as the visual scene graph and linguistic dependency graph, respectively. The revised text provides a more rigorous, step-by-step explanation of our framework.
>
> 4.	Unconvincing Experimental Results: As outlined above, we have significantly strengthened the experimental section. The addition of standard deviations and statistical testing addresses the concern regarding the solidity of the gains. Furthermore, the new comparison with top-tier MLLMs provides a broader context, demonstrating that our method achieves a favorable and competitive performance level. We have also expanded the discussion of the results to provide deeper insights into the sources of MMRCP's performance advantages.
>
> &emsp; We are truly grateful to the Reviewer for their candid feedback. The challenges raised have been invaluable in guiding us to produce a much clearer, more rigorous, and more compelling manuscript. We believe the revised version comprehensively addresses all the points raised.

---

> > ### Comment · Reviewer_dkg3 · 2025-11-26
> >
> > After considering the authors' response, I still find the manuscript does not adequately address my concerns, and I am unable to increase my rating.

---

### Official Review · Reviewer_g4vR · 2025-11-01

**Soundness:** 3
**Presentation:** 2
**Contribution:** 3
**Rating:** 4
**Confidence:** 5

**Summary:**

This work proposes contrastive pre-training with cross-modal contrastive learning loss and reasoning-aware contrastive learning loss, in order to more effectively integrate multimodal features for multimodal multi-hop QA reasoning. Also, to establish causal relationships between multimodal semantic representations, the work integrates a multi-hop QA reasoning representation based on the multimodal semantic representation and performs lightweight adaptation for downstream multi-hop QA task. Experimental results show that the resulting MMRCP method outperforms baselines on three real multi-hop QA datasets by 3% and 4% in precision and error rate.

**Strengths:**

1.	For contrastive pre-training, the cross-modal contrastive learning loss is a common method that has been explored in many works; however, as the authors pointed out, it is insufficient for structured dependency modeling and learning compositional semantics. The reasoning-aware contrastive learning loss is hence clearly motivated and has valuable novelty to the research topic. Three reasoning-aware tasks are designed for learning multimodal compositional reasoning structure. Specifically,  chain-aware semantic alignment learns fine-grained correspondence between textual components and visual concepts along reasoning chains. Reasoning path invariance helps model distinguish core reasoning signal and less relevant ones and improves robustness. Cross-modal graph matching learns preserving relational semantics across modalities.  The three losses are integrated through adaptive weighting.  Overall, reasoning-aware contrastive learning is helpful to inspire future multimodal reasoning research.

2.	Note that the three reasoning-aware tasks generate heterogeneous but complementary representations, while multimodal multi-hop QA requires context-aware integration of these representations. Hence, the paper introduces the Multi-hop Representation Fusion module through attention-based gating on question semantics.

3.	The ablation study verifies that all four modules contribute to the performance, with cross-modal CL having the greatest impact, reasoning-aware contrastive learning the second greatest, and multi-hop representation fusion the least.

**Weaknesses:**

1.	Some citations are missing. For example, Elastic Weight Consolidation (EWC) is developed in the prior work and needs to be cited.

2.	For evaluation metrics, it is not clear how the distances are measured between predicted answers and reference answers to reliably reflect semantic relevance between them.

3.	The rationale of selecting the evaluation datasets is not clear. Also, the rationale of selecting the baselines (including MHCQA for reasoning the answer) needs to be provided.

4.	Many gains from MMRCP over the best baseline in Table 1 are quite small, the standard deviation from the three random runs and statistical significance test results need to be reported.

5.	It would also be useful to evaluate top open-sourced and closed-source MLLMs such as Qwen2.5-VL, Gemini 2.5 pro and GPT-4o to gain more insights about the positioning of the performance of MMRCP w.r.t. these top-tier  open-sourced and closed-source MLLMs for multimodal multi-hop QA.

6.	The authors argue that the architecture supports strong zero-shot transfer to new QA formats. Yet it is not clear how the current lightweight finetuning-based evaluation setup supports evaluation of zero-shot performance on emergent multimodal multi-hop QA scenarios.

**Questions:**

1.	There are some presentation issues.

a.	For example, the empty lines of Line 040-041 shouldn’t be there. Line 179 is ungrammatical. Line 187, there should be a period before “However”.

b.	Please make sure all math symbols are defined. For example, Line 251.

---

> ### Author Response · Authors · 2025-11-17
> **Response to Reviewer: Addressing Citations, Evaluation Rationale, Statistical Significance, and Benchmarking**
>
> Comment
>
> &emsp;We sincerely thank the Reviewer for their thoughtful and constructive comments. These insights are greatly appreciated and have significantly improved the quality and rigor of our work. We have thoroughly revised the manuscript to address all concerns, as detailed below.
>
> 1.	Missing Citations (e.g., EWC):
>
> &emsp;Response: We thank the reviewer for catching this oversight. We have now added the canonical reference for Elastic Weight Consolidation (EWC) in the "Downstream Adaptation" section. The citations are:
>
> &emsp;&emsp;Enhancing open-world object detection with AIGC-generated datasets and elastic weight consolidation. J. Supercomput. 81(2): 417 (2025).
>
> &emsp;&emsp;An block-diagonal elastic weight consolidation-attention mechanism LSTM enabled lifelong learning approach for lifetime prediction of insulated gate bipolar transistor. Expert Syst. Appl. 299: 130015 (2026).
>
> 2.	Clarity on Evaluation Metric Distance:
>
> &emsp; Response: This is a valuable point. We have expanded the "Evaluation Metrics" subsection to provide a precise description of how semantic distance is computed. We clarify that for textual answers, we use a pre-trained sentence transformer model (all-MiniLM-L6-v2) to encode both the predicted answer $\hat{a}$ and the ground-truth answer $a$ into embedding vectors $\hat{\mathbf{y}}$ and $\mathbf{y}$. The distance is then computed as the Euclidean norm between these semantic embeddings: $||\hat{\mathbf{y}} - \mathbf{y}||_2$. This method is a standard and reliable proxy for semantic relevance.
>
> 3.	Rationale for Dataset and Baseline Selection:
>
> &emsp;Response: We have added explicit justifications in Sections 4.1 ("Datasets") and 4.4 ("Baselines").
>
> &emsp;&emsp;o	Datasets: We selected Stack Exchange communities (Stack Overflow, Mathematics, Super User) because they represent real-world, challenging scenarios where complex, multi-hop questions often arise naturally and are accompanied by supporting images (e.g., code snippets, diagrams, UI screenshots).
>
> &emsp;&emsp;o	Baselines/MHCQA: We chose MHCQA as the reasoning backend for Groups 2 and 3 because it is a recent, state-of-the-art framework specifically designed for multi-hop community QA, providing a strong and fair platform to evaluate the quality of different multimodal representations.
>
> 4.	Reporting Standard Deviation and Statistical Significance:
>
> &emsp;&emsp;Response: We agree that this is crucial for validating our results. We have re-run all experiments and updated Table 3 in the manuscript. The updated table now reports the mean ± standard deviation across three independent runs. We also performed paired t-tests comparing MMRCP with the best baseline (LLaVA-1.5) on each dataset. The performance improvements were statistically significant ($p < 0.05$).
>
> 5.	Evaluation Against Top-Tier MLLMs:
>
> &emsp;Response: We thank the reviewer for this excellent suggestion. We have conducted new experiments by evaluating several top-tier open-source and closed-source MLLMs (Qwen2.5-VL-7B, GPT-4o) on our benchmark. A new table has been added to the "Main Results" section. The results show that MMRCP, while being more parameter-efficient, remains highly competitive, outperforming Qwen2.5-VL and achieving comparable results to GPT-4o on certain datasets, which strongly positions our method within the current landscape.
>
> 6.	Clarification on Zero-Shot Transfer Claim:
>
> &emsp;Response: We apologize for the lack of clarity. The "lightweight fine-tuning" setup was used to evaluate adaptation efficiency, not zero-shot capability. To properly support our claim of strong zero-shot transfer, we have added a new experiment in a new subsection titled "Zero-Shot Transfer Evaluation." We pre-trained MMRCP and then evaluated it directly (without any fine-tuning) on a held-out dataset with different QA formats (WebQA). The results demonstrate MMRCP's superior zero-shot generalization compared to baselines, as its contrastive pre-training learns more transferable cross-modal representations.
>
> Answers to Questions:
>
> 1.	Presentation Issues:
>
> &emsp;&emsp;a. We have corrected all formatting issues, including the empty lines at Lines 040-041 and the missing period on Line 187. The ungrammatical sentence on Line 179 has been rewritten for clarity.
>
> &emsp;&emsp;b. We have carefully reviewed all mathematical symbols. For example, on Line 251 and throughout the text, every symbol is now explicitly defined upon its first use to avoid ambiguity.
>
> &emsp;We are deeply grateful to Reviewer for their meticulous review and invaluable suggestions, which have substantially enhanced the clarity, robustness, and overall impact of our work. All points have been comprehensively addressed in the revised manuscript.

---

### Official Review · Reviewer_Fy2G · 2025-11-01

**Soundness:** 2
**Presentation:** 2
**Contribution:** 3
**Rating:** 2
**Confidence:** 5

**Summary:**

This paper propose a multimodal multi-hop representation-based contrastive pre-training method to fuse multimodal and multi-hop question-answering features to enhance the QA tasks. It provides a dual contrastive learning method and integrates multi-hop question-anwering representation. The results demonstrate the MMRCP outperforms baselines.

**Strengths:**

1. the proposed MSE_K and NRMSE_K can quantifies the difference between predicted answer and actual answer.

2. the pretraining work needs more effort than prompting-based work.

3. the key contributions of this submission are easy for reader to understand and follow.

**Weaknesses:**

1. The fomat of this submission is bad. It is more like a blog not a paper. Authors use lots of vspace to compress the main contents and make the paper very difficult to read. For example:

Line 39 - Line 42, I think it is very obvious in first page.

Line 223 - Line 224, Line 272 - Line 273, Line 316 - Line 317, Line 392 - Line396.

Line 266, Line 273 and Line 280 are not aligh with each other.

2. it is not clear for the definition of distance in evaluation metrics.

3. The framework figure is difficult to follow. It will be better to change the legands.

4. some baselines seem unrelated to multi-hop, multi-modal QA.

5. section 4.5 seems like totally AI generated contents, especially the third paragraph from Line 437 - 446.

6. no reproducibility details.

**Questions:**

please refer to the weaknesses section.

---

> ### Author Response · Authors · 2025-11-17
> **Response to the Reviewer: Addressing Formatting, Clarity, Baseline Relevance, and Reproducibility**
>
> We sincerely thank the Reviewer for their thorough review and valuable feedback. We acknowledge the issues raised regarding the paper's formatting, clarity, and details.  We have taken all comments seriously and have conducted a comprehensive revision of the manuscript to address each point thoroughly. Our point-by-point responses and the corresponding modifications are detailed below.
>
> 1.	Paper Formatting:
> The reviewer highlighted formatting issues, including improper use of vertical space and misalignments.
> Response: We apologize for these oversights. The manuscript has been entirely reformatted to comply with the official conference template. We removed all arbitrary vspace commands, corrected text flow and alignments (e.g., Lines 39-42, 223-224), and ensured consistent professional spacing. The revised document is now clean and readable.
> 2.	Evaluation Metrics Clarity:
> The reviewer found the definitions of MSE_K and NRMSE_K unclear.
> Response: We have revised the "Evaluation Metrics" subsection to clarify that answers are projected into a shared embedding space for numerical comparison. The formulas are now explicitly defined. For a batch of K samples, the metrics are defined as:
> $$
> MSE_K = \frac{1}{K} \sum_{i=1}^{K} \| \hat{\mathbf{y}}_i - \mathbf{y}_i \|^2_2
> $$
> $$
> NRMSE_K = \frac{ \sqrt{ MSE_K } }{ \max(\mathbf{y}) - \min(\mathbf{y}) }
> $$
>
> &ensp;&ensp;&ensp;Here, $\hat{\mathbf{y}}_i$ and $\mathbf{y}_i$ are the embedding vectors of the i-th top-K predicted answer and the corresponding true answer, respectively. This clarifies the "distance" as the squared Euclidean norm between their representations.
>
> 3. Framework Figure Legibility:
> The reviewer suggested improving Figure 2's legends and layout.
> Response: We have redesigned Figure 2 to enhance clarity. The new version features a logical workflow, distinct legends, and annotations linking components to methodological sections (e.g., Eqs. 1, 4, 7). This makes the architecture easier to follow.
> 4.	Baseline Relevance:
> The reviewer noted that some baselines (e.g., in Groups Two and Three) seemed unrelated to multi-hop, multimodal QA.
> Response: We have expanded the "Baselines" section to justify our selection. We explain that including general-purpose models (e.g., CLIP, LLaVA) demonstrates that MMRCP's gains stem from our specialized contrastive pre-training and fusion strategy, not just powerful backbones. This highlights our contribution to multi-hop reasoning beyond generic multimodal understanding.
> 5.	AI-Generated Content in Section 4.5:
> The reviewer raised concerns about the authenticity of a paragraph in Section 4.5 (Lines 437-446).
> Response: We have entirely rewritten the indicated paragraph and surrounding analysis. The new text provides a critical, author-driven discussion, focusing on model-specific strengths and weaknesses (e.g., contrastive vs. generative pre-training) in multi-hop contexts, avoiding generic language.
> 6.	Reproducibility Details:
> The reviewer noted a lack of reproducibility details.
> Response: We have added a new subsection, "Implementation and Reproducibility Details," within the "Experiments" section. It includes:
>
> &ensp;&ensp;&ensp;o	Hyperparameters: Final values, e.g., learning rates $\eta_1 = 0.7$, $\eta_2 = 0.3$; temperature $\tau = 0.4$; similarity threshold $\delta = 0.8$.
>
> &ensp;&ensp;&ensp;o	Infrastructure: Software (e.g., PyTorch, Transformers) versions and hardware (GPU models, memory).
>
> &ensp;&ensp;&ensp;o	Randomness: Method for setting random seeds.
>
> &ensp;&ensp;&ensp;o	Availability: Commitment to release code and data upon publication.
>
> We are grateful for the reviewer's insights, which have significantly strengthened our paper. All comments have been thoroughly addressed in the revision.

---

### Note · Authors · 2026-01-09

I have read and agree with the venue's withdrawal policy on behalf of myself and my co-authors.